# Converting UML-based ontology conceptualizations to OWL with Chowlk⋆

Serge Chávez-Feria[0000−0002−7454−9202],
Raúl García-Castro[0000−0002−0421−452X], and
María Poveda-Villalón[0000−0003−3587−0367]

Ontology Engineering Group, Universidad Politécnica de Madrid, Spain
serge.chavez.feria@upm.es, {rgarcia, mpoveda}@fi.upm.es

**Abstract.** During the ontology conceptualization activity, developers usually generate preliminary models of the ontology in the form of diagrams. Such models drive the ontology implementation activity, where the models are encoded using an implementation language, typically by means of ontology editors. The goal of this demo is to take advantage of the developed ontology conceptualizations in order to accelerate the ontology implementation activity. For doing so we present Chowlk, a converter to transform digital UML-based ontology diagrams into OWL. This system aims at supporting users in the generation of the first versions of ontologies by reusing the ontology conceptualization output.

**Keywords:** Ontology engineering · ontology conceptualization · Ontology implementation

## 1  Introduction

One important step in ontology development is the conceptualization one, during which the ontology development team defines a set of concepts and properties to represent the knowledge of an specific domain. Often, this conceptualization is materialized in a diagram that displays the ontology elements and their connections. From this model, the ontology implementation is carried out normally using an ontology editor, encoding the model in OWL. In this process the diagram is in most of the cases only used as a guideline to manually implement the ontology. To address this issue, some tools have been proposed to allow the graphical creation or modification of ontologies[5, 2, 4, 3].

The presented work aims at leveraging the above-mentioned diagrams in order to allow for a smoother transition from the conceptualization activity to the actual implementation, that is transforming XML diagrams following a UML-based ontology visual notation into OWL. For doing so, rather than building a graphical ontology editor, the process builds on top of well-adopted system as diagrams.net which allows collaborative and synchronous edition of the conceptualization models.

---

⋆ This work has been supported by the BIMERR project funded from the European Union's Horizon 2020 research and innovation programme under grant agreement no. 820621.

## 2    Chowlk features

Chowlk is a web application that takes as input an ontology conceptualization created with diagrams.net and generates the OWL implementation. The service is available online[1] (See Figure 1) and the source code is shared in a GitHub repository[2] under the Apache 2.0 license.

The conceptualization should follow the Chowlk visual notation[3] which is also provided as a diagrams.net library[4] to allow users to easily reuse the correct shapes to avoid problems during the transformation and save to time during the conceptualization.

The converter is able to identify concepts, object properties, datatype properties, and restrictions between those elements. Also, the converter identifies ontology metadata, namespaces and the prefixes being used in the model, due to specific blocks dedicated to this type of information. Labels to each ontology element are added during the detection process. Once the XML diagram is loaded into the system, Chowlk starts searching for all the ontology elements in the conceptualization ignoring shapes not included in the specification. After the detection and creation of the corresponding associations between the ontological elements, Chowlk proceeds to write the implementation using the OWL language. Finally, the ontology is provided in two downloadable formats: Turtle and RDF/XML. (See Figure 2).

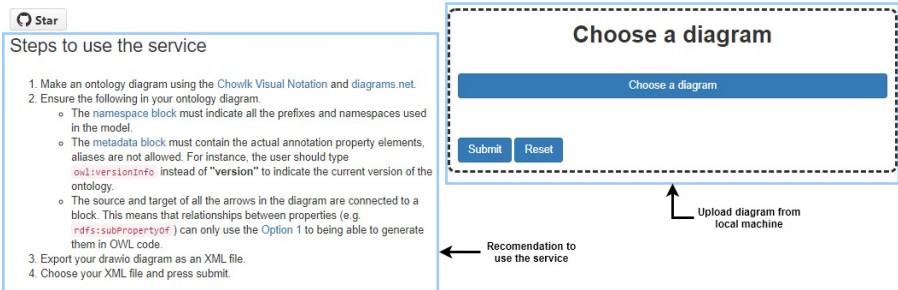

Fig. 1: Screenshot excerpt of Chowlk home page

---

[1] https://chowlk.linkeddata.es

[2] https://github.com/oeg-upm/Chowlk

[3] https://chowlk.linkeddata.es/chowlk_spec

[4] https://github.com/oeg-upm/chowlk_spec/blob/master/
chowlk-drawio-library.xml

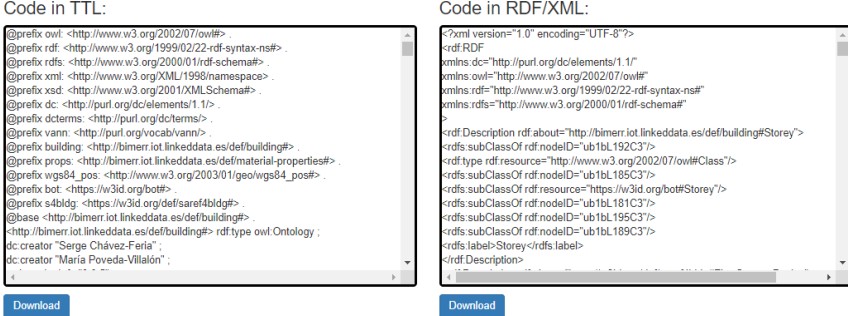

Fig. 2: Screenshot of the web GUI showing the output serializations.

## 3   Architecture

The architecture of the Chowlk converter can be shown in Figure 3. It was developed as a web application that takes as input an ontology conceptualization, created with diagrams.net, and the Chowlk visual notation to generate the OWL implementation. Figure 3 also shows the modules composing the system, namely: the detection module, the association module, and the writing module. The detection modules identifies all the building blocks represented in the diagram that follow the Chowlk visual notation, discarding any shape that does not correspond to the notation ones. This detection is done by analyzing the attributes of the XML data structure. The association module performs the connection between the ontological elements found in the detection module. This association is done by taking the classes of the ontology as the core elements. For each class, the module iterates over the object properties and datatype properties found in the conceptualization and performs the association based on the proximity of the shapes. Finally, the writing module takes all the ontology elements detected in the previous steps and serializes them in RDF, along with the respective restrictions between them.

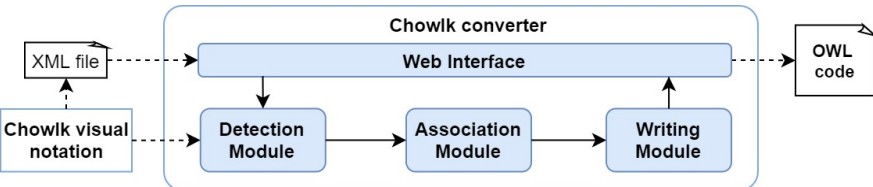

Fig. 3: Chowlk architecture.

## 4    Demonstration

During the demonstration,[5] some examples will be developed to show the partic-
ipants how to use Chowlk. First, the conceptualization will be developed using
diagrams.net following the Chowlk visual notation. The shapes for the construc-
tion of the model will be provided as a diagrams.net library that can be loaded
into the diagramming tool. The final conceptualization will be exported an up-
loaded into the Chowlk web app that will generate the corresponding OWL
implementation. The GUI will return the ontology in two formats: RDF/XML
and Turtle. As a final step, the ontology will be uploaded into Protégé to demon-
strate the syntax correctness of the exported implementation.

## 5    Conclusions and future work

This paper presents the Chowlk converter that facilitates the ontology develop-
ment process by leveraging the ontology conceptualization diagrams to transform
them into OWL. The system is currently being used within the authors research
group[6] and also by researchers from other institutions[7]. Next steps involve the
announcement and promotion of the system to get feedback from users. Addi-
tionally, we plan to develop a REST API so that Chowlk can be easily integrated
within third-party software such as OnToology [1]. It is also planned to develop
a plugin for diagrams.net in order to embed the ontology generation feature into
the diagramming tool. Besides, we plan to integrate the prefix.cc[8] service to sug-
gest an URI for prefixes not declared. Including support for including additional
metadata, such as comments and labels, is also being explored.

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
