# OpenReview forum: "Converting UML-based ontology conceptualizations to OWL with Chowlk"
_eswc-conferences.org/ESWC/2021/Conference/Poster_and_Demo_Track — ESWC2021 P&D_

### Official Review · AnonReviewer4 · 2021-04-14
**Demo of a system (in development)**

**Rating:** 7
**Confidence:** 3

**Review:**

The paper briefly introduces the current system (Chowlk) of a tool that turns XML diagrams into OWL using a set of visual notations. It is not entirely clear what these notations are, and the paper could be improved by better explaining what the different components of the architecture do. The paper also links to a demo video that demonstrates the idea and the uses of the system.

I think that the community could benefit from using tools like this, as it offers a more general method to quickly build ontologies without knowledge about Protege or OWL descriptions, but based on the vague description of the system I believe that the system is still in development and could use the strength of having the results from the user-studies that is mentioned in passing in the text.

Dependent on the number of high-quality system demos, the paper could also be presented as a poster, rather than a demo.

**Anonymity:**

Yes, I would like my review to remain anonymous.

---

### Official Review · AnonReviewer1 · 2021-04-14
**Cool and polished idea, translation implementation could use a little work**

**Rating:** 6
**Confidence:** 3

**Review:**

The demo presents an approach to OWL generation from UML diagrams based on the online diagramming software diagrams.net, a translation mechanism that ingests such diagrams, and an accompanying vocabulary for how the diagrams are to be expressed in order to be parseable by the translator.

The idea of reusing established UML modeling techniques is a good one; this opens up ontology engineering to a much broader class of software engineer than it has traditionally been available to. The vocabulary is logically constructed and seems (though see question below) sufficiently fully featured in terms of the OWL constructs allowed.

Some drawbacks from my point of view:

1) The dependence on this particular online tool rather limits the usability of the approach outside of initial prototyping and mockups. It would be much nicer if one could build a fully featured workflow around this promising idea.

2) I haven't had time to dig deep into this, but it would be interesting to know if the vocabulary supports all OWL constructs, and if not, which are left out. A sentence or two in the paper and in the Chowlk spec would suffice.

3) When I attempted to run the translation tool, only the absolutely simplest examples worked without flaw. Trying anything more complex resulted in translation failures, even though the spec was followed (as far as I can tell) exactly. It was very frustrating to get an error message without any details, download a "faulty" diagram, load it up and see some edge being marked in red, but that red edge being compliant with the published Chowlk spec. Especially since the constructs were dragged-and-dropped from the provided Chowlk shapes palette. Could the error messages maybe be made more expressive? This was consistent enough and irritating enough to warrant a reduction in scoring; if it were resolved, I'd gladly give a 7 score ("Good paper, accept") instead.

Minor issue: Section 3, second to last sentence, indicates that proximity is used to associate shapes. I assume this is a language mistake -- surely, per the spec, the connections (through edges) should be used for such association?

**Anonymity:**

Yes, I would like my review to remain anonymous.

---

### Official Review · AnonReviewer2 · 2021-04-16
**Chowlk: A useful tool for OWL modelling**

**Rating:** 8
**Confidence:** 3

**Review:**

This paper introduces a demo which will present Chowlk a converter to transform digital UML-based diagrams to OWL.

Modelling support that helps to simplify the construction of OWL ontologies is very timely and welcome, as it both reduces the modelling effort, and increases the quality (as automation helps to avoid human errors).

The design of the converter seems sound to me, the tool in online and the source-code provided, which makes it easily useable for both occasional users and more interested experts.

Finally, the paper is well written, so this is a very good demo for the conference, and I recommend accepting it.

I am not giving the paper a higher grade because of my own uncertainty, as I cannot guarantee the novelty of the tool given that I am not fully aware of the state of the art.

(After having read the other reviews, I would like to mention a danger of the approach: given that UML as a modelling language is rather inexpressive, unless I am very much mistaken, usage of the tool might restrict the usage of other features of OWL. It would be interesting for future research to study how the chosen input model and transformer could be extended to encourage users to go beyond the restricted expressivity. This is only a suggestions, though, and does not effect my judgement of the paper.


**Anonymity:**

Yes, I would like my review to remain anonymous.

---

### Decision · Program_Chairs · 2021-04-19

Accept